

# Opinion: Inferring Process from Snapshots of Cloud Systems

Graham Feingold[1], Franziska Glassmeier[2], Jianhao Zhang[1,3], and Fabian Hoffmann[4]

[1]National Oceanic and Atmospheric Administration (NOAA), Chemical Sciences Laboratory, Boulder, Colorado, USA
[2]Department of Geoscience & Remote Sensing, Delft University of Technology, Delft, Netherlands
[3]Cooperative Institute for Research in Environmental Sciences (CIRES), University of Colorado, Boulder, Colorado, USA
[4]LMU, Munich Germany

**Correspondence:** Graham Feingold (graham.feingold@noaa.gov)

**Abstract.** The cloudy atmospheric boundary layer is a complex, open, dynamical system that is difficult to fully characterize through observations. Aircraft measurements provide cloud dynamical, thermodynamical, and microphysical properties along a flightpath, at high spatial/temporal resolution (order 10 m/0.1 s). These data are essentially contiguous "snapshots" in time of the state of the cloud and its environment. Polar-orbiting satellite-based remote sensing yields snapshots of retrieved cloud and aerosol properties once or twice a day at spatial scales on the order of 250 m, but these are usually averaged to scales of ≈ 20 – 100 km to reduce uncertainties. Neither approach tracks a parcel of air in time, a view that would yield more direct insights into the evolving system. Nevertheless, our long experience with aircraft and satellite-based remote sensing has taught us much about atmospheric processes, suggesting that one can gain insights into processes from these snapshots. Using mostly previously published work we present examples of collections of observation snapshots that reveal various degrees of process-level understanding. We couch the discussion in terms of the concepts of space-time exchange, ergodicity, and process vs. observation timescales. It is our hope that this paper will encourage the atmospheric sciences community to explore the value of these concepts more deeply.

## 1 Introduction

The atmospheric system, like many other complex, open systems comprises myriad coupled processes, a very large number of coupled geophysical variables (GVs), and a huge number of degrees of freedom. The atmospheric system is thus notoriously unpredictable (e.g., Bauer et al., 2015; Selz et al., 2022). Survival instincts have for millennia driven humans to observe and record the weather, with ever-increasing levels of sophistication, especially over the past century and a half. Current observational systems include highly sophisticated surface-based in-situ and remote sensing instruments, aircraft-borne cloud and aerosol microphysics probes, and passive and active remote sensing systems in space. Aircraft measurements provide cloud dynamical, thermodynamical, and microphysical properties along a flightpath at high spatial/temporal resolution (order 10 m/0.1 s). At typical aircraft speeds of 50 - 100 m/s these measurements quickly become spatially de-correlated, which limits what one can learn about the evolution of a process based on consecutive measurements. Thus these data are essentially snapshots in time of the state of the cloud and its environment. Satellite-based remote sensing yields snapshots of retrieved cloud and aerosol properties at spatial scales on the order of 250 m, but these are usually aggregated to scales of ≈ 20 –





100 km to reduce uncertainties. The aggregation creates further challenges to addressing processes, particularly when process timescales are short.

    Ideally the study of processes would track the system as it evolves in space and time, i.e. from a Lagrangian rather than an Eulerian perspective. This is particularly true if one seeks a causal relationship between a perturbation of one variable on the state of the system, e.g., cloud response to aerosol perturbations. Given the speed of most of our measurement platforms

and the rate of movement of atmospheric systems this generally proves to be challenging. The notion that one can infer time-evolving 'process' from static 'snapshots' can be demonstrated through the question of whether one can learn the rules of football or chess from infrequent snapshots of the game, and even from different occasions on which the game is being played. (We will assume that the rules of the game are invariant.) Atmospheric observing systems provide glimpses of the state of the system on different days, and under different large-scale meteorological conditions. Since cloud systems are highly sensitive to

meteorological conditions this can be viewed as a game in which the rules are changing. Stratification to similar meteorological conditions becomes essential but there remains a question of the importance of the history of the evolving system on its way to the observation point, i.e. stratification might have to be extended to matching air parcel/mass histories. Thus even in a data-rich world, understanding how best to use data to improve our understanding of the atmosphere requires thought.

    Ergodicity is a concept introduced in the late 19th century that has its origins in the study of systems in equilibrium by

the statistical physics community (Boltzmann, 1884). Ergodicity relates to the idea of a system characterized by a motion in phase space that, given long enough, fills the entire space. The concept of ergodicity conveys the idea that the average state of the system can be equally characterized by either the collection of all the system states obtained when following one specific system over time, or, alternatively, and equivalently, by a suitably sampled collection of individual realizations of the system at any given time. Consider the challenge of sampling a system of moving gas molecules in a confined volume. One approach

would be to track a single molecule with some temporal frequency – say 1000 times – to understand the mean state of the system. Another might be to take an instantaneous snapshot of 1000 spatially separated molecules, well spread out over that space in the same closed system to deduce these processes. (In reality, for a gas, the number of samples and molecules would be much larger - on the order of an Avogadro's number of molecules for the system to be in thermodynamic equilibrium.) The system is considered ergodic if the mean state of the system based on these two approaches is the same, i.e., if a system

is ergodic, the dynamical description (tracking a single molecule) can be replaced by a much simpler probabilistic view (a snapshot of 1000 molecules). Loosely speaking this can be thought of as a (phase-) space-time exchange: the average of the properties of many spatially *separated* particles at one time ('space') is equivalent to the average of the same properties of one molecule sampled over many times ('time'). Newer developments offer that the system does not have to fill the entire space, and pose the idea of ergodicity weak enough to hold for a given system, and strong enough to have significant consequences

(see e.g., Ashley, 2015). We will not attempt to cover the huge body of literature that has accumulated on various aspects of ergodicity; instead we will stick to a much simpler, conceptual level that ties to a sample of questions in the atmospheric sciences.

    A simple example of a system that is ergodic is the rolling of a die: Whether 100 people role a die once, or one person rolls a die 100 times, the probability of obtaining a given number, say 3, is the same ($1/6 \times 100 = 16.67\%$). The counter example





might be that if one person engages in Russian roulette 6 times or six people engage in Russian roulette once (each with their

own firearm), the outcome will be very different – i.e the system is decidedly non-ergodic.

In a social sciences context one can think of tracking one person 1000 times over a period of interest to study a given

phenomenon vs. sampling 1000 persons at different stages of this same phenomenon but at one point in time (Hunter et al.,

2024). (In this case 'space' refers to the separate individuals.) In economic theory the goal is to understand how well expected

values of economic metrics compare with time averages. For example, because economics systems are typically far from steady

state, an individual investor might question the relevance of the present-day growth of a collection of investment portfolios

('space') to the projected growth of their personal portfolio ('time') (e.g., Peters, 2019).

A key component of ergodic systems is that the 'rules of the game' need to be the same. Thus the dice used in the above

example cannot be weighted to bias the outcome differently between repeated rolls of the dice. In the social sciences, ergodicity

becomes much harder to achieve because the systems involve people who are not identical and might behave unpredictably –

i.e., the rules of the game are not uniform across the population. In atmospheric systems, meteorology is a key determinant of

how a system evolves. Meteorological changes thus equate to changes in rules of the game, which depend on the state of the

atmosphere, the particular attributes of the system being studied, and how far they are from steady state.

In addressing the ergodicity of a system, the magnitude of the process timescale of the system process(es) $\tau_{\mathrm{proc}}$ relative to

the observation timescale $t_{\mathrm{obs}}$ is a helpful organizing metric. The Deborah number (Reiner, 1964) is defined as

$$\mathcal{D} = \frac{\tau_{\mathrm{proc}}}{t_{\mathrm{obs}}}. \tag{1}$$

If the duration over which the process is observed is long enough to detect the characteristic timescale of the process under

investigation then $t_{\mathrm{obs}} \gg \tau_{\mathrm{proc}}$ and $\mathcal{D}$ is small – e.g., a radar tracking a convective storm system over its lifecycle allows one

to study the microphysical processes (minutes) associated with the precipitation generated by the storm over the course of its

lifetime (hours). Systems for which $\mathcal{D} \gg 1$ evolve slowly enough to render them essentially static to the observer. An example

of the latter is the human observation of geochemical erosion of rocks. For a select process, a $\mathcal{D} \ll 1$ coveys the idea that the

observation time is long enough to allow the system to fully explore its state-space, a necessary (but insufficient) condition for

ergodicity. Conversely, if the observation time is too short for full exploration of the state space then the system is non-ergodic.

Observation timescales in the atmospheric sciences range from minutes to multiple days to decades, depending on the analysis

approach, lifetime of the platform, and perspective. (As will be discussed below, a time snapshot does not necessarily imply an

observation period approaching zero.) The determination of the Deborah number becomes especially interesting for systems

that feature processes on multiple temporal and spatial scales (Bossen and Mauro, 2024), which translates into various possible

$\tau_{\mathrm{proc}}$ that could be dominating the evolution captured in the data. Clouds with processes that range from the microscale of cloud

microphysics (seconds to minutes) up to the large-scale evolution of cloud-controlling factors (days) are a prime example of

such a multiscale system (Alinaghi et al., 2025a, b).

A related concept is Taylor's frozen turbulence hypothesis (Taylor, 1938). If one's goal is to measure the characteristics

of a turbulent eddy, the only practical way to do so is to place a sensor in the medium and allow the eddies to advect past a

sensor. Taylor's "frozen" hypothesis assumes that the statistical properties of turbulent eddies do not evolve significantly as they



advect past the sensor. In other words one assumes that the eddy is close to steady over the observation period, or $\tau_{\text{proc}} \ll t_{\text{obs}}$.
A space-time exchange follows naturally; e.g., if $\tau_{\text{proc}} \ll t_{\text{obs}}$ then a time-height plot from a vertically pointing radar at a fixed ground site can be converted to a distance-height plot using knowledge of the mean wind speed.

Below we explore whether process information can be gleaned from snapshots of cloud systems with a number of examples, most of which are previously published results from the peer-reviewed literature. We cast results in terms of ergodicity, space-time exchange, and observation and process timescales that determine the Deborah number. The paper therefore revisits well-
known results in a framework that we believe will benefit the field – both conceptually and practically. We do not weigh in on whether our systems are ergodic in the strict sense. We end with a discussion of results and a perspective on implications for the field of aerosol-cloud-climate interactions.

## 2 Examples

In a multiscale system, the Deborah number will depend on the timescale of the process of interest $\tau_{\text{proc}}$ as well as the tools
at our disposal to make the observations. As we survey work that has shown some success in space-time exchange, we will present examples that study process and observation timescales that fulfill $\mathcal{D} \ll 1$ for a process of interest. We will distinguish between two main types based on the relative magnitudes of the meteorological timescale $\tau_{\text{met}}$, i.e., the timescale at which the 'rules of the game' change:

1. Type 1: A single snapshot for which meteorological conditions are essentially constant (invariant 'rules of the game');
$\tau_{\text{proc}} \ll t_{\text{obs}} \ll \tau_{\text{met}}$

2. Type 2: A composite of snapshots for which meteorological conditions vary (variant 'rules of the game'); $\tau_{\text{proc}} \ll \tau_{\text{met}} < t_{\text{obs}}$.

### 2.1 Drop effective radius profiling (Type 1)

Cloud drop effective radius ($r_e = \int r^3\, n(r) dr / \int r^2\, n(r)\, dr$, the ratio of the third to the second moments of the drop size
distribution), is an important GV that is strongly tied to the radiative properties of a cloud. $r_e$ can be retrieved using passive radiometry in the near infrared (e.g., Nakajima and King, 1990), but given the nature of passive measurements, and the strong weighting to cloud-top, it is challenging without hyperspectral measurements to directly profile $r_e$ over the depth of warm clouds (King and Vaughan, 2012). These profiles are desirable because they enable inferences on dominant cloud processes such as condensation or collision-coalescence. Rosenfeld and Lensky (1998) proposed a method to generate Temperature-$r_e$
(T-$r_e$) profiles by sorting as a function of temperature the $r_e$ at the tops of individual cumulus clouds within a single satellite scene. (Here temperature is a proxy for height since the relationship between the two can be fairly easily established.) In other words, the authors looked at data from the same time stamp and used the spatial variability within the cloud scene to generate T-$r_e$ profiles (Fig. 1). The fundamental idea is that these individual, spatially separated clouds are exemplars of the $r_e$ profile in a single local cloud of arbitrary height, provided the meteorology in the field is approximately constant. In essence this means





that 'space' (individual clouds separated from one another) and 'time' (temporal evolution of an individual cloud –i.e., process) are equivalent.

    Ruiz-Columbié (2003) pointed out that the profiling method proposed by Rosenfeld and Lensky (1998) invokes ergodicity – the first reference we have found to ergodicity in cloud systems. Lensky and Rosenfeld (2006) tested this idea by analyzing multiple scenes using geostationary satellite retrievals. Individual cloud cells were tracked to study the temporal evolution of

cells. In addition, the standard profiling method as described above, was applied in the different scenes. The authors were then able to demonstrate an "exchangeability between time and space".

    Setting aside technical difficulties associated with satellite-based measurements in broken cloud fields, Zhang et al. (2011) tested the space-time exchange by analyzing the output of large eddy simulation (LES) of shallow trade-wind cumulus clouds (Xue and Feingold, 2006) assuming no retrieval error. They demonstrated that to a surprising degree of precision, the profile of

$r_e$ composited from the tops of individual, spatially distributed clouds follows that of individual clouds. This manifestation of some level of ergodicity in convective cloud fields is, in our minds, quite profound since it implies an internal self-consistency of cloud development within a cloud field immersed in an approximately constant thermodynamic state. Over the course of the day, the clouds influence their environment as they transport moisture and evaporate. Apparently the effect of clouds on their environment is sufficiently spatially homogeneous such that subsequent cloud fields still obey the space-time exchange for $r_e$

profiling.

    To put this example in the perspective of the Deborah number, we interpret $t_{\mathrm{obs}}$ not as the instantaneous snapshot in time, but as the time represented by the evolution of the full suite of clouds in the scene. This includes several cycles of everything from the smallest nascent clouds through to the deepest, developed clouds, and the small decaying clouds – yielding a $t_{\mathrm{obs}}$ on the order of several hours. As noted above, this self-consistency between cloud-evolution cycles would not hold in the presence of

meteorological gradients across the cloud field (i.e., for $t_{\mathrm{obs}} > \tau_{\mathrm{met}}$). For example, in the presence of a gradient, cloud A on one side of the domain would be growing at a different rate and experiencing different interactions with its environment than cloud B on the other side of the domain, in which case one would not *a priori* expect the members of population of clouds to evolve in the same way. In this way 'constant meteorology' equates to 'constant rules of the game'. The relevant process timescale $\tau_{\mathrm{proc}}$ is associated with the evolution of an individual cloud or, more specifically, the height increment between the individual

clouds, ordered by height (say 100 m). For typical updrafts on the order of meters per second $\tau_{\mathrm{proc}}$ would be on the order of minutes. Together, this results in $\tau_{\mathrm{proc}} \ll t_{\mathrm{obs}} \ll \tau_{\mathrm{met}}$ (Type 1). The underlying reasons for the success of $r_e$ profiling are in our minds not well understood. Lensky and Rosenfeld (2006) have argued for the "constant renewal of growing cloud tops with cloud bubbles that replace the cloud tops with fresh cloud matter from below". Examining a single cloud, Bretherton and Smolarkiewicz (1989) describe the buoyancy difference between a cloud and its undisturbed surroundings in terms of spreading

gravity waves that rapidly equilibrate the buoyancy gradient via compensating downdrafts. The rate of this equilibration is much faster than the entrainment-mixing timescale. These gravity waves are associated with entrainment (detrainment) regions when the cloud buoyancy relative to the environment is increasing (decreasing) with height. This homogenization of the buoyancy in the cloud field might explain why cloud bubbles across the cloud field experience similar histories. Moreover, the rapidity of the equilibration suggests that the $r_e$ profiling would not be dependent on the detailed nature of the slower entrainment-mixing



(inhomogeneous vs. homogeneous; Baker et al., 1980). The underlying reasons for the success of $r_e$ profiling would clearly benefit from a deeper investigation.

## 2.2  Compositing within a single snapshot (Type 1)

The next example derives from modeling studies, although the phenomenon has since been tested through observations. A recent approach to investigating the nature of complex cloud systems using LES model output is through the creation of
composites of select variables within a snapshot of a single cloud field evolving under steady meteorological conditions. Bretherton and Blossey (2017) studied the aggregation of mesoscale patches of higher humidity in shallow cumulus fields. By compositing the model output based on quartiles of column integrated total water mixing ratio they were able to elucidate the processes that lead to cloud clustering, in the absence of external forcing of such pattern and even without precipitation and radiation. The study by Janssens et al. (2022) achieves the same effect as compositing by scale filtering, and highlights the
intrinsic nature of moisture aggregation. A similar analysis for the case of shallow mesoscale overturning circulations (SMOCs) was published by Narenpitak et al. (2021). See also Janssens et al. (2022) and George et al. (2023) for additional modeling and observational perspectives, respectively.

Zhou and Bretherton (2019) applied the methodology to non-precipitating stratocumulus (Sc). We will walk through this example more carefully to put it into the perspective of the prior example of space-time exchange. Consider a single Sc cloud
scene exhibiting a regular closed-cellular structure. Sample this scene at various points across the domain and sort the columns by their total water path (TWP). Then composite these columns, ordering them in bins of TWP. Probe the LES output to obtain the dimensions of the cell and the flow-field in these TWP bins. Following this procedure reveals a self-sustaining mesoscale circulation with a horizontal scale of $\approx 20 - 30$ km in a boundary layer of $\approx 1$ km depth. The circulation is characterized by a weak updraft at the core and thickest part of the of the Sc. Descending free tropospheric air diverges at cloud top and moves
air along the shallow cloud-top slope from moister regions to drier regions. The circulation is reinforced by the horizontal gradient in cloud-top radiative cooling between cell core and edge. Cold, dry air penetrates down into the boundary layer at the cell edges and convergence of air from the edges towards the cell core completes the circulation. (Fig. 2). Thus, an archetypal Sc cell, derived from composite sampling of spatially disparate parts of the system fits well with our general understanding of closed-cellular convection. Being based on a model that resolves the relevant processes, it also lends itself to a deeper
mechanistic understanding of these cells; e.g., when the horizontal gradient in cloud-top radiative cooling between cell core and edge is removed, the circulation is weaker (Zhou and Bretherton, 2019).

Here too the success of this approach is, in our minds, profound: Composited variables are used to show the typical behavior of a system based on *percentiles of those variables drawn from non-contiguous parts of the domain*. Similar to the case of T-$r_e$ profiling, one can successfully use spatially separated 'fragments' (percentiles) of the broader cloud field to build the general
characteristics of a Sc cell. Closed-cellular convection is an atmospheric analog of Rayleigh-Bénard convection. Over limited domains (order 100 km), boundary conditions such as sea surface temperature and subsidence are approximately constant, as





is the depth of the boundary layer. Cell aspect ratios are approximately 30:1. Under these homogeneous conditions there is enough self-consistency within the Sc for a space-time exchange to be useful.

Considering the Deborah number analysis for this case, $t_{\text{obs}}$ amounts to the number of stratocumulus cells over the duration of the simulation, similar to the number of cloud cycles considered in the previous example. Even when restricting simulation time to less than an hour to avoid variability within the diurnal cycle, one will still sample several cells, resulting in $t_{\text{obs}} \sim$ hours. In contrast, the process timescale $\tau_{\text{proc}}$ is related to the eddy turnover time of about 20 min. With $\tau_{\text{proc}} \ll t_{\text{obs}}$ and $\tau_{\text{met}} \gg t_{\text{obs}}$ (fixed meteorology) we consider this example as Type 1.

## 2.3 Autoconversion (Type 2 and Type 1, depending on the methodology)

The initiation of precipitation via collision-coalescence is a topic of great interest to cloud physicists and climate modelers. Collision-coalescence is commonly separated into the self-collection of cloud droplets less than about 40 $\mu$m in diameter ('autoconversion') and the collection of small cloud droplets by larger raindrops ('accretion'). Stephens and Haynes (2007) presented a means to quantify autoconversion timescales using satellite-based passive (visible and infrared wavelengths) and active (3-mm radar) measurements. These respectively yield cloud optical depth COD, effective radius $r_e$, and radar reflectivity $Z$ ($\propto \int r^6 n(r) dr$, the 6th moment of the drop size distribution $n(r)$). Central to this idea is that the Moderate Resolution Imaging Spectroradiometer (MODIS)-derived COD and $r_e$ provide information on the cloud droplet mode of the drop size distribution while the radar-derived $Z$ is highly sensitive to larger drops. The method is facilitated by the relative robustness in the modal diameter of the cloud droplets, even as the concentration decreases as a result of autoconversion. (Naturally the system is more complicated if one includes other cloud processes like advection, sedimentation or drop breakup in a dynamically evolving system.) The framework for the Stephens and Haynes (2007) retrieval is a simple theoretical model of the continuous collision-coalescence process (Bowen, 1950), plus an assumption of the collection kernel (Long, 1974). Some manipulation of the equations yields

$$P \cdot h = c_1 \frac{2}{3} \rho_w r_e \text{ COD } \bar{Z} H[\bar{Z} - Z_c], \tag{2}$$

where $P$ is the collision-coalescence rate, $h$ the column depth over which $P$ is measured, $\bar{Z}$, a mean cloud-averaged radar reflectivity, $H$, the Heaviside function, expressing a sampling of all radar data for which $\bar{Z} >$ some critical value $Z_c$, and $c_1$ is a function of the collection kernel and has units of m$^{-3}$ s$^{-1}$. (For details see Stephens and Haynes 2007.)

To demonstrate the approach, the authors used MODIS retrievals of $r_e$ and COD, together with CloudSat $\bar{Z}$ to derive all terms on the right hand side of the equation. Data were collected over oceans between 60N and 60S for June, July, and August 2006, and importantly were limited to -15 dBZ $< \bar{Z} < 0$ dBZ in order to emphasize the initiation of rain formation via autoconversion. Integrated coalescence rates $P \cdot h$ [g m$^{-2}$ s$^{-1}$] were then plotted as a function of liquid water path LWP ($\propto r_e \cdot$ COD) [g m$^{-2}$] such that the ratio of the ordinate to the abscissa represents a timescale for autoconversion. Figure 3 reproduces Figure 3 in Stephens and Haynes (2007) where one sees a rather broad scatter of the points but that the majority (73%) of the measurements fall between a timescale of 26 min and 3 h.



This example raises a number of interesting points. In keeping with our theme, it utilizes snapshots of data to infer process
understanding (autoconversion timescales), in this case constrained by a simple model. A question that arises is why the
process rates are relatively poorly constrained. In part we believe that this is due to the larger number of degrees of freedom
in the system (e.g., a cloud plus a rain mode). Other reasons might include the fact that a column of air might include drops
that have been advected into the column, or that the data sorting (-15 dBZ $< \bar{Z} < 0$ dBZ) is not rigorous enough to exclude
accretion; in other words, the data might reflect processes not considered in the simple model. Or even if the data is dominated
by autoconversion, Eq. (2) might be too simple.

From a measurement perspective, the cloud and rain components from which the autoconversion rate is derived are separated
spatially and temporally because the radar is on a different satellite trailing the $r_e$ and COD measurements by a few minutes.
The use of multiple GVs derived from different instruments with different view volumes might also be a problem. Moreover,
data from many different cloud scenes are aggregated. These suggest a limit to how well the autoconversion timescale can be
quantified with this approach.

For this example, the GVs used in the retrieval are sensitive to the full range of drop sizes, with $\tau_{\text{proc}}$ the time it takes for
drops to grow from newly formed droplets to raindrops, i.e., on the order of 15 – 20 minutes. Because the data derive from
many different conditions, the observation timescale $t_{\text{obs}}$ is on the order of many days, during which the meteorology changes
($\tau_{\text{met}} < t_{\text{obs}}$). We therefore categorize this as Type 2 because the process under consideration (autoconversion) is obscured
by the varying large-scale meteorological conditions. Thus, the uncertainties in the derived timescales are likely a result of
changing rules of the game as well as because of uncertainties associated with the retrieval methodology. Depending on the
goals of the study and quantification requirements, such analyses may still be useful.

Building on the success of $r_e$ profiling presented in section 2.1, we consider another approach to quantifying autoconversion
and accretion, this time in the form of process rates. Taking advantage of new passive remote sensing instruments such as the
Research Scanning Polarimeter (RSP; Alexandrov et al., 2015) and the HyperAngular Rainbow Polarimeter (HARP; McBride
et al., 2024) that provide information on cloud top effective variance $\nu_e$ as well as cloud top $r_e$, we explore whether the added
information on cloud-top $\nu_e$ can quantify these collection rates. By definition

$$\nu_e = \frac{\int (r - r_e)^2 \, r^2 \, n(r)dr}{r_e^2 \, \int r^2 n(r) \, dr}$$

which reduces to

$$\nu_e = \frac{\int r^2 \, n(r)dr \int r^4 \, n(r)dr}{(\int r^3 n(r) \, dr)^2} \, - \, 1.$$

Some rearrangement yields

$$r_{43} = \frac{\int r^4 \, n(r)dr}{\int r^3 \, n(r) \, dr} = (\nu_e + 1)r_e,$$

which is an effective drop size with more weight on higher order moments than $r_e$. Since both cloud-top $r_e$ and cloud-top $\nu_e$
can be retrieved, we explore whether this higher weighting can provide information on drop collection. First we apply the $r_e$
profiling technique to $r_{43}$ using the same bin microphysics LES output used by Zhang et al. (2011) to demonstrate that cloud
top measurements of $r_{43}$ provide very similar profiling capabilities to $r_e$ (Fig. 4). This means that, ignoring remote sensing





uncertainties, one can retrieve $r_e(z)$ as well as $r_{43}(z)$. Next we calculate the collection rates of the drop size distributions associated with this bin microphysical model output and separate them into autoconversion and accretion rates. Figure 5 a, b show that autoconversion rates increase robustly with $r_{43}$ but then begin to level off and decrease at some larger $r_{43}$. In con-

trast, accretion rates increase steadily with increasing $r_{43}$, as expected, dominating the collection process once drops become sufficiently large (Fig. 5 c, d). Of note is that high values of $r_{43}$ are not always a strong constraint on autoconversion and accretion rates because large $r_{43}$ may derive from low values of liquid water mixing ratio $q_c$ co-occurring with low values of $N$. Nevertheless the self-consistency exhibited by shallow cumulus convection – at least in terms of the manifestation of how ratios between moments evolve – provides insights beyond the retrieval of GVs themselves. The uncertainties are large, but

typical of those associated with modeling (e.g., Khairoutdinov and Kogan, 2000). Whether this approach is useful will require more rigorous testing for a broad range of conditions.

As in the case of the T-$r_e$ profiling this approach can be considered Type 1 since $\tau_{\mathrm{met}}$ is infinitely large (it is constant across the scene of interest), $\tau_{\mathrm{proc}}$ is on the order of 15-20 minutes, and $t_{\mathrm{obs}}$ is on the order of hours.

## 2.4   Liquid water path – Drop concentration relationships (Type 2 or Type 1 depending on the methodology)

LWP and drop concentration ($N$) are GVs that characterize the bulk properties and microphysical structure of a cloud system and are central to aerosol-cloud interactions. Much has been written on the influence of aerosol particles on $N$, and the resultant effects on cloud albedo $A_c$, as well as LWP and cloud fraction $f_c$. Twomey (1977) considered aerosol effects on $A_c$ without the complications of adjustments to LWP and $f_c$ but more recently these adjustments have been shown to be critical to the evaluation of aerosol-cloud-climate forcing (Bellouin et al., 2020). A satellite-based view of a very large number of snapshots

of cloud systems in LWP-$N$ space exhibits an interesting separation into two branches (e.g., Gryspeerdt et al., 2019), often referred to as an "inverted V" (Fig. 6a). These have been interpreted as the low $N$, precipitating branch 1 where LWP increases with $N$ as a result of precipitation suppression and the high $N$, non-precipitating branch 2 where LWP decreases with $N$ as a result of aerosol-related enhancement in droplet evaporation. Note that the interpretation of these diagrams presupposes a relationship between state-space sampling and processes, which is not a priori justified. Nevertheless these arguments are not

without merit given our understanding of aerosol-cloud interactions in detailed (typically single case) large eddy simulations (LES). For example, Xue et al. (2008) showed a similar inverted V in $f_c$ -$N$ diagrams, and the role of $N$-dependent droplet evaporation has been elucidated by Wang et al. (2003), Ackerman et al. (2004), and Bretherton et al. (2007).

But can we strengthen this intuition that a compilation of snapshots does indeed inform us about processes? Glassmeier et al. (2019) and Glassmeier et al. (2021) analyzed large ensembles (order 130) of LES to explore the evolution of cloud

systems covering a large range of boundary layer temperature and water vapor profiles. The output of these nocturnal LES in LWP-$N$ space shows a remarkable convergence of cloud systems towards a steady-state line that also exhibits an inverted V shape and is qualitatively similar to the satellite snapshots. Hoffmann et al. (2020) examined the same large ensemble of LES to address the extent to which the LWP-$N$ diagram represents physical processes captured by the LES. By applying mixed layer theory to the individual LES the authors showed a breakdown of LWP tendencies ($d\mathrm{LWP}/dt$) due to radiation, surface





fluxes, evaporation, precipitation, and cloud top motion. The topographical maps in LWP-$N$ space show the clear dominance of individual processes in the very regions that they are expected (Fig. 6b). Evidence for process understanding from inverted Vs can also be found in Zhou et al. (2025) where lead-lag analysis is applied to satellite data to help establish causal relationships between $N$ and LWP or LWP and $N$. Of note is that during the daytime, solar heating has been shown to have a flattening effect on the negative LWP-$N$ slope using large ensembles of diurnal LES (Zhang et al., 2024; Chen et al., 2024b).

The view based on the synthesis of these large ensembles of LES has recently been augmented by a heuristic model based on equations representing basic microphysical processes such as evaporation and collision-coalescence, and thermodynamical cloud water recharge (Hoffmann et al., 2024). The model also generates inverted Vs and explores the model parameters that change the slopes of the two branches and how these relationships are affected by stochastic perturbations. By varying the timescale $\tau_{\mathrm{prt}}$ and thus importance of these perturbations, the heuristic model clarifies the role of different internal processes

(Fig. 7). For infrequent perturbations (Fig. 7, b), the evolution remains dominated by the deterministic mesoscale processes as discussed above. Here $\tau_{\mathrm{proc}}$ is a mesoscale timescale of about 10 h, which is $\ll t_{\mathrm{obs}}$ (multiple days in these simulations) such that $D \ll 1$, enabling the long-time evolution of the stochastic system to ergodically explore its steady state. This ergodicity implies that snapshots would equally sample this steady state and as the steady state is characterized by balancing processes, this in turn implies a tight relationship between processes and snapshots. For the most frequent perturbations (Fig. 7, c), however, the

stochastic component of the evolution might dominate, resulting in $\tau_{\mathrm{proc}} = \tau_{\mathrm{prt}} \ll t_{\mathrm{obs}}$. In this situation, the sampling reflects the stochastic external forcing rather than the internal processes of the mesoscale evolution of the cloud field. We still have $D \ll 1$ and a relationship between snapshot and process but the dominant process is the stochastic perturbation. The heuristic model also explores the effect of external variability and shows that the shape and position of the inverted V changes with the large-scale conditions. Such external variability blurs the relationship between snapshots and mesoscale cloud processes, similar to

the effect of stochastic perturbations. The effect of the external variability can be interpreted as variability in meteorological conditions in aggregated satellite-based LWP-$N$ data, suggesting that a composite based on many different meteorological conditions will be associated with uncertain slopes.

   Against this conceptual background, we classify LWP-$N$ analysis as performed by Gryspeerdt et al. (2019) as Type 2 for the following reasons: $\tau_{\mathrm{proc}}$ is on the order of 10 h (Chen et al., 2024a), and the compositing of large numbers of satellite snapshots

means that $\tau_{\mathrm{proc}} \ll t_{\mathrm{obs}}$ (order many days), meeting the requirement that $\mathcal{D} \ll 1$. However with $t_{\mathrm{obs}} > \tau_{\mathrm{met}}$, meteorological variability will limit the ability of this type of analysis to quantify cloud processes.

   Goren et al. (2025) go so far as to argue, that cloud-field processes are completely dominated by large-scale processes, i.e., $\tau_{\mathrm{proc}} = \tau_{\mathrm{met}}$ amounts to the timescale of large-scale evolution. In this case the inverted V is simply a consequence of a spatial change in cloud depth as one follows the prevailing winds off the coast of a Sc-capped area out over warmer waters, together

with aerosol-meteorological co-variability. Initially the boundary layer is shallow and characterized by high $N$ because of proximity to continental aerosol sources. As one moves southwestward over the warmer ocean, the boundary layer deepens, allowing for higher LWP. At the same time, $N$ decreases because of increasing distance from aerosol sources. This describes the negative LWP-$N$ branch. At some point clouds become thick enough to precipitate, which decreases $N$ – thus defining the positive branch. To calculate the Deborah number from the perspective of Goren et al. (2025) analysis, $\tau_{\mathrm{proc}}$ is on the order of





 days and, as a result of compositing, $t_{\mathrm{obs}}$ is many days. Again, meteorological variability obscures the process ($t_{\mathrm{obs}} > \tau_{\mathrm{met}}$) and we classify this analysis as Type 2.

Zhang et al. (2022) and Zhang and Feingold (2023) offered an alternative by analyzing LWP-$N$ at a 20 km footprint-level within $1° \times 1°$ boxes under approximately constant meteorological conditions. In this case quantitative information on microphysical processes is likely more reliable ($t_{\mathrm{obs}} \ll \tau_{\mathrm{met}}$; Type 1).

The analysis of Goren et al. (2025) relates slopes to *boundary layer processes* as opposed to the *microphysical process* view of Gryspeerdt et al. (2019) or Zhang and Feingold (2023). Both are valid and both suggest a connection between snapshot and process. The study by Possner et al. (2020) might be considered an attempt to disentangle the two contributions by controlling for boundary-layer depth. Thus the interpretation of inverted Vs requires thorough reflection on the processes that are included/resolved by the temporal observation timescale.

**2.5 Snapshots plus reanalysis (Type 1)**

A variation on the examples of a snapshot that yields information on the cloud field is an approach that combines multiple snapshots with meteorological reanalysis data to study temporally-evolving systems, thus more directly addressing causality. (The study of Goren et al. (2025) discussed above falls into this category.) Gryspeerdt et al. (2021) used thousands of MODIS snapshots from the Aqua satellite, reanalysis wind fields, and ship emission information to assess the temporal evolution

of cloud responses to ship emissions. The use of satellite snapshots to unveil the temporal dimension in cloud adjustment provides a better sense of process, especially given the multi-hour timescale associated with LWP and $f_c$ adjustments to aerosol perturbations. More recently Murray-Watson et al. (2023) studied cold-air outbreaks at higher latitudes by spatially and temporally matching wind trajectories based on reanalysis-derived meteorological fields with MODIS retrievals of cloud properties. While the use of reanalysis data in these examples is helpful, the methodology still relies on compositing, which

inevitably increases $t_{\mathrm{obs}}$ to multiple days, meaning that meteorological confounders cannot be avoided.

Zhang et al. (2025a) took a pseudo-Lagrangian approach to studying cloud systems that uses the GOES-16 geostationary satellite retrievals of $r_e$ and COD (and derived LWP) but avoids compositing. We show here one example that exemplifies this technique. A single GOES snapshot of a cold-air outbreak with a resolution of ~3 km over the western North Atlantic Ocean is the start point. 1000 hPa winds from ERA-5 reanalysis at the snapshot time are used to generate *instantaneous trajectories*,

along which GOES cloud micro- and macro-physical retrievals are extracted. In contrast to conventional Lagrangian tracking of cloud evolution in a series of geostationary satellite images, this *instantaneous trajectory* approach invokes 'space-time exchange' and allows one to infer time-evolving microphysical *and* boundary layer processes in one single GOES snapshot. Here the justification for the 'space-time exchange' is that the large-scale meteorological conditions are evolving much more slowly than the rate of cloud evolution, i.e., the timescale of changes in the 'rules of the game' is much larger than the

timescale of target processes. In this case the dominant timescale of variability is that of cloud evolution, which is often the case for marine cold-air outbreaks. A plot of LWP vs. $N$ shows a trace that contains valuable information about the underlying boundary layer and microphysical processes along the trajectory (Fig. 8). Figure 8 also shows a diagram that identifies the





effect that processes have on the directionality of the trace. These include microphysical processes that are strongly influenced by the large-scale forcing, e.g., the SST gradient: (i) drop activation – increases in LWP and $N$ (arrow 1); (ii) condensational

growth – an increase in LWP at constant $N$ (arrow 2); (iii) collision-coalescence – a decrease in $N$ at constant LWP (arrow 3); (iv) precipitation and evaporation – a sink for $N$ and LWP (arrow 4); and (v) entrainment-mixing – a reduction in $N$ and LWP, with directionality depending on whether the mixing is homogeneous or inhomogeneous (arrows 5.1 and 5.2). While the approach does not provide an unambiguous parsing of processes, it does provide process fingerprinting over the spatial dimension of a snapshot and demonstrates the usefulness of space-time exchange in process inference.

Considering the Deborah number for this example, an *instantaneous trajectory* spans more than 10 degrees, equivalent to an $t_{obs}$ of ∼12 hours for a boundary layer wind-speed of 20 m s $^{-1}$. This allows one to consider both microphysical processes ($\tau_{proc} \sim$ minutes) and boundary layer deepening processes ($\tau_{proc} \sim$ hours). $\mathcal{D}$ is less than 1 for both of these process timescales. On the other hand, the rate of change in SST spatial gradient is on the order of days and thus not ergodically sampled such that variability is dominated by cloud processes. Thus, this example is considered as Type 1.

## 3    Discussion

We now discuss a number of topics that emerge from the examples shown above.

### 3.1    Gleaning causality from state-space diagrams

The sampling of one system at multiple times is the case study/timeseries approach that is frequently applied in the atmospheric sciences. The reason for its popularity may be historical since it can be accomplished with fewer resources. Based on

a case study – either modeling or observations – one might show a clear relationship between Y and X, e.g., the deepening of clouds (Y) in response to an aerosol perturbation (X). With time, such responses tend to become the cornerstones of our understanding, but often without sufficient focus on the fact that the result derives from one case. The case study approach stands in contrast to the use of large samples of data that attempts to generalize the results for a wide range of conditions. The development of networks of surface observations, satellite-based remote sensing, and regional models has pushed the field

towards generalization. For the aerosol-cloud deepening example given above, sampling many cases under different conditions provides generality in terms of responses but makes it far more difficult to attribute the deepening to aerosol. The large data sets or model ensembles contain many different meteorological conditions that might themselves be driving the deepening, with aerosol manifesting as a confounding factor. Some have argued for more focus on understanding the co-variabilities between system variables – an approach that seeks to expose the confounders (Mülmenstädt and Feingold, 2018). Others stratify the

large data sets by the variables considered most likely to explain the response, hoping that the stratification is fine enough to be sure that results are robust. Machine learning is also proving useful as a means of teasing out counterfactual conditions (see e.g., Zhang et al., 2025b; Chen et al., 2022, — would these clouds have changed in response to a perturbation had the meteorological conditions been the same?). As discussed elsewhere (e.g., Harte, 2002; Feingold et al., 2016), an iterative consideration of the time-oriented (case study) and (phase-)space-oriented (attempt to generalize) approaches is essential to solidifying our





understanding. Ergodic thinking and space-time exchange therefore lies at the interface of the case-study/generalization inter-
face.

## 3.2 The geostationary satellite view

With the new generation of geostationary satellites that produce time derivatives of GVs from advanced imagers and radiome-
ters, it can be argued that we can observe processes more directly, e.g., (Christensen et al., 2020), so why focus on snapshot
data? The program of record comprises a wealth of relatively untapped atmospheric data from polar orbiting satellites that can
be mined for insights. Aircraft data also fall into this category. Exploring ways to maximize the information content of these
data is therefore worthwhile.

While Type 1 data applications don't necessarily require geostationary satellite data, more frequent snapshots of the same
scene provide the opportunity to compare geostationary retrievals of the evolving cloud state with inferences about processes
based on polar orbiting satellite retrievals, albeit based on instruments with different data characteristics, resolution, and preci-
sion. In principle this could be a more rigorous test of the space-time exchange.

The geostationary point of view might be especially valuable when accompanied by knowledge of the meteorology (e.g.,
from reanalysis). For example, the study of aerosol-cloud interactions would be able to discern temporal changes in cloud
fields in response to aerosol perturbations (response time on the order of 10 minutes) as well as adjustments in LWP and cloud
fraction that have timescales on the order of 10 h (Chen et al., 2024a; Glassmeier et al., 2021) – within the context of the
evolving meteorology. At high solar zenith angles (SZA) retrievals are more problematic but events that lie within the optimal
SZA window (less than 65°; Grosvenor et al., 2018) will be valuable. As already noted, extant studies have utilized composites
of satellite snapshots to construct temporal evolutions of aerosol-cloud interactions in ship-tracks (Gryspeerdt et al., 2021). By
enabling higher resolution temporally contiguous observations of a given cloud process, the geostationary view should provide
more confidence in the causal nature of the process.

## 4 Summary and Outlook

In an era of increasingly large volumes of atmospherically-relevant data, we have addressed the question of how much one
can learn about atmospheric processes from infrequent snapshots of the state of the system. It is common in the atmospheric
sciences to aggregate large samples of instantaneous 'snapshots' of data – usually in the form of geophysical variables (GVs) –
and attempt to infer knowledge of the underlying processes that produce relationships between the GVs. To quote Mülmenstädt
and Feingold (2018) "..temporally evolving system[s] with an inherent memory [are] studied with a Markovian, 'snapshot- in-
time' methodology, which assumes that processes are related to the current state of the system, and have no memory of past
states." In doing so we inherently assume causal relationships between these variables and ignore the spatiotemporal unfolding
of the system with its inherent timescales. Here we have attempted to provide some perspectives on this approach through the
use of commonly applied GV state diagrams. The framework for discussion is the statistical physics concept of ergodicity,
which we apply *ad hoc* rather than adhering to its manifold and more rigorous definitions.





A central concept of ergodicity is the exchange of 'space' and 'time'. For example, one may attempt to relate a snapshot of spatially separated parts of a system ('space') to the temporal evolution of a part of that system ('time') and in so doing infer knowledge of 'process', which is inherently causal. Aircraft in-situ measurements and satellite-based remote sensing retrievals tend to be used in this way since there is no way to track the evolution of the relevant (process-dependent) small volumes of the atmosphere with time. (An exception in recent times is the use of geostationary satellite systems with their new and improved GV retrieval capabilities.) Missions such as the A-Train of satellites (Stephens et al., 2018) were conceived to provide statistics of the state of the system through retrievals of GVs. The temptation to equate statistical correlation with causation is at the heart of this paper and the existence of huge volumes of high quality data does not obviate the need for a deeper look at methodology.

The examples shown here point to some of the opportunities and limitations of the use of snapshots beyond a broad statistical analysis of geophysical variable space. They serve to demonstrate examples of whether and which processes (cloud scale or large scale) one might infer from snapshots within the (here) loosely defined framework of ergodicity and space-time exchange and with the added perspective of Deborah number analysis. We have deliberately avoided detailed discussion or rigorous adherence to the many aspects of ergodicity and instead focused on its chief derivative, *space-time exchange*, as applied to real-world studies of atmospheric systems. The Deborah number ($\mathcal{D}$ = (process timescale) / (observation period)) is a useful way of quantifying whether the observation period is long enough for the system to have fully explored its state space with respect to a certain process scale.

Our examples have been characterized into two distinct types of data categories, the primary difference being the relative magnitude of the observation period and the timescale of meteorological variability:

1. Type 1: $\mathcal{D} \ll 1$, $\tau_{\mathrm{proc}} \ll t_{\mathrm{obs}} \ll \tau_{\mathrm{met}}$. 'Space' is associated with different and separated cloud elements *within the same scene* and 'time' is associated with the evolution of a single cloud element. The two examples that seem to show the most promise are the profiling of $r_e$ in cumulus and the compositing of closed cellular stratocumulus. In both of these examples, spatially separated parts of the cloud field are composited in ways that allow the evolution of an individual component to be analyzed. The ability to constrain meteorology such that a cloud process and its timescale dominate variability in the data is likely key to the success of this approach.

2. Type 2: $\mathcal{D} \ll 1$, $\tau_{\mathrm{proc}} \ll \tau_{\mathrm{met}} < t_{\mathrm{obs}}$. The large composites of data associated with Type 2, including climatological studies, translate to the inclusion of a large range of meteorological conditions, or changing 'rules of the game'. Examples are autoconversion of cloud droplets to rain (Stephens and Haynes, 2007) and LWP-$N$ analyses after Gryspeerdt et al. (2019). The changing meteorology likely limits our ability to quantify autoconversion timescales and the slopes of the inverted V in the LWP, $N$ analyses. Careful stratification of the data by similar conditions might help to improve quantification (e.g., Zhang et al., 2022).

An interesting case is the LWP-$N$ state-space since both Type 1 and Type 2 classification has been arrived at. Hoffmann et al. (2024) used a heuristic model that considers simple equations for microphysical processes such as evaporation, collision-coalescence, and rain formation, as well as thermodynamic cloud-water recharge and entrainment to show that inverted V shapes emerge quite naturally. For fixed meteorology (no perturbations), the timescale analysis suggests Type 1. On the other





hand, Goren et al. (2025) considered LWP-$N$ in a Lagrangian sense; they composited trajectories as they move away from the west coasts of continents and transition from closed-cell stratocumulus to broken cumulus to precipitating conditions, and showed that they also manifest inverted Vs. In this view, the inverted V is described as driven primarily by meteorology and co-variability of meteorology with aerosol. The timescale analysis indicates Type 2 as a result of the compositing of the data.
Both views hold merit since different processes and timescales are being targeted.

The use of additional information afforded by meteorological reanalysis is an interesting variation on the 'process-from-snapshot' efforts. The analysis of LWP-$N$ evolution in cold air outbreaks (section 2.5) is an example of how ancillary meteorological data describing a trajectory through such a system can be used together with LWP and $N$ retrievals to glean information on both boundary layer and microphysical processes. The success of this example rests on the fact that the cold-air outbreak
event changes much more slowly than the winds advected through the scene. The approach is similar to Gryspeerdt et al. (2021), and Murray-Watson et al. (2023) in that boundary layer winds are used to advect the system, but is fundamentally different in that that our proposed approach uses a single snapshot instead of a large composite, thus avoiding the meteorological confounders.

While most of the results presented here derive directly from previously published work, an extension of the T-$r_e$ profiling
proposed by Rosenfeld and Lensky (1998) has been explored. We have shown that similar to the usefulness of $r_e$ profiling demonstrated by measuring $r_e$ at the tops of spatially separated clouds in the cloud field, new remote measurements of cloud top drop spectral variance $\nu$ might allow one to retrieve both $r_e$ and $r_{43}$ (the ratio between 4th and 3rd moments of the drop size distribution). The latter is shown here to have the potential to provide useful constraints on autoconversion and collection rates.

In closing, we hope that exploring the concept or ergodicity, or its more accessible 'space-time exchange' will lead to better understanding of how snapshot data (and large collections thereof) are useful for inferring process understanding. By examining the ratio of process timescales to the duration of observation, the Deborah number provides a useful way to quantify how well a system can explore its state space over the period of observation for a given process, and the extent to which meteorology might confound quantification by changing the rules of the game.

We proffer that strict definitions of ergodicity may not be necessary for snapshots to be useful for understanding processes but encourage the community to dig much deeper into ergodicity in atmospheric systems – as has occurred in the fields of statistical physics, economics, and social sciences. When doing so, a primary concern should be the extent to which the physical drivers (e.g., meteorology and aerosol) or the 'rules of the game' are consistent. Thus we advocate for targeting processes that are conducive to full sampling of the state space for selected observational periods, avoiding compositing of data derived from
different meteorological conditions. Finally, we note that while in-situ measurements and remote sensing retrievals continue to improve, practical limitations may get in the way, e.g., our ability to retrieve remotely $r_e$ in small, broken clouds. Challenges posed by retrievals should not prevent us from refining our conceptual thinking of how to derive better understanding from large data sets.



*Author contributions.* GF conceived the project and developed the ideas with FG and FH. JZ added ideas and new analysis. GF wrote the
first draft. All authors contributed to writing and editing.

*Competing interests.* Two of the authors are associate editors of ACP.

*Acknowledgements.* We thank Prof. Huiwen Xue for generating the model output analyzed in Figs. 4 and 5. The model output was originally
published in Xue and Feingold (2006).

    GF and JZ acknowledge support from the NASA ACTIVATE program under Reimbursable Agreement number NNL23OB04A.
FG acknowledges support from The Branco Weiss Fellowship Society in Science, administered by ETH Zurich and by the Eu-
ropean Union (ERC, MesoClou, 101117462). Views and opinions expressed are however those of the author(s) only and do
not necessarily reflect those of the European Union or the European Research Council Executive Agency. FH acknowledges
support from the Emmy Noether program of the German Research Foundation (DFG) under grant HO 6588/1-1.



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





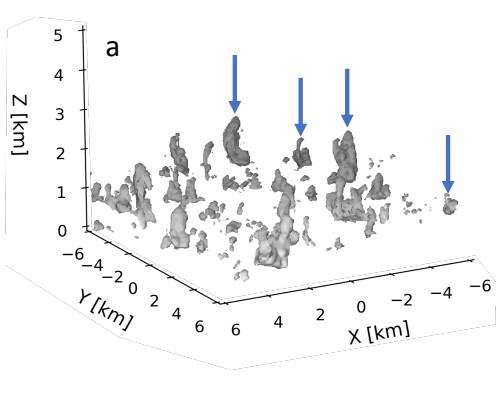

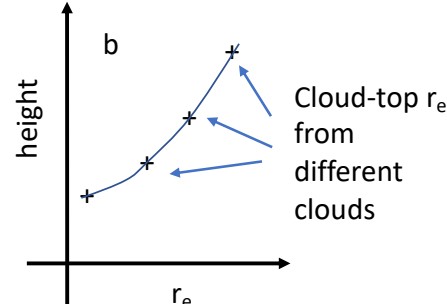

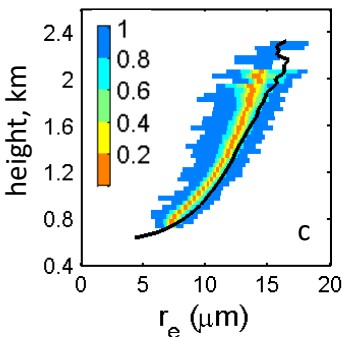

**Figure 1.** (a) Schematic of the Rosenfeld and Lensky (1998) method of profiling of drop effective radius $r_e$ (T-$r_e$); (b) The cloud top $r_e$ of individual clouds in the scene are composited to form a T-$r_e$ profile; (c) Testing of the method using LES output of shallow cumulus clouds. Solid line: domain-mean profile of $r_e$; colored contours: $r_e$ based on individual cloud tops. Colors indicate percentiles. Reproduced from Fig. 1e in Zhang et al. (2011), courtesy of ACP.



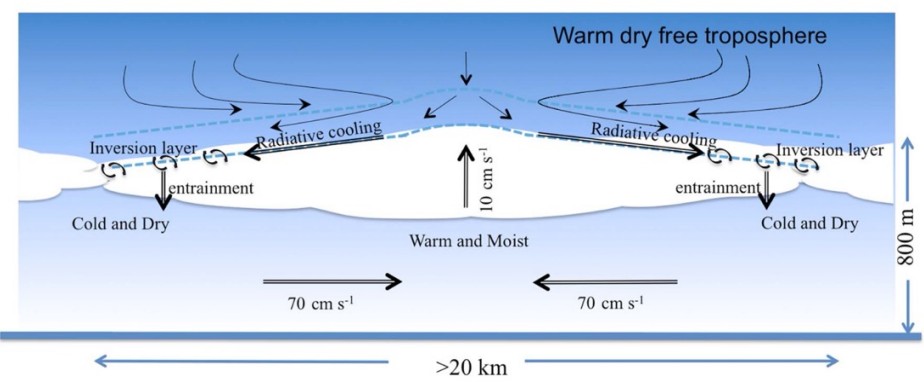

**Figure 2.** Schematic describing the processes associated with a typical stratocumulus cell. The schematic is based on the compositing of spatially separated parts of cells in a LES-generated cloud field. (Fig. 15 in Zhou and Bretherton (2019), courtesy of Wiley.)



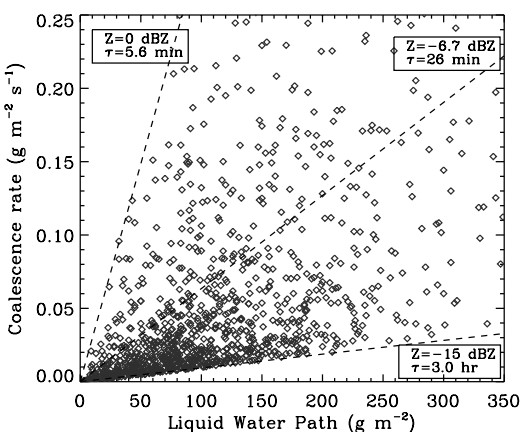

**Figure 3.** Autoconversion timescales ($\tau$) based on satellite-based radar and spectrometer data using Eq. (2). Autoconversion timescales derived from the slope of Eq. (2) vary between 5.6 min and 3 h, with 73% of data between 26 min and 3 h. (From Stephens and Haynes (2007) Fig. 3, courtesy of Wiley, with permission.)



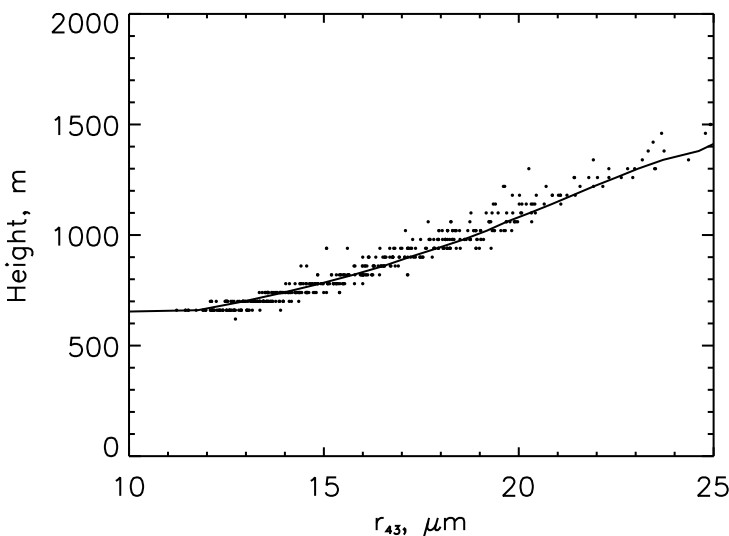

**Figure 4.** Profiling of $r_{43}$ – the ratio of the 4th to the 3rd moments of the drop-size distribution – following the method of Rosenfeld and Lensky (1998). Data points are derived from the tops of individual clouds within a field of cumulus using the LES output in Zhang et al. (2011). The solid line represents the mean profile based on all clouds in the field.



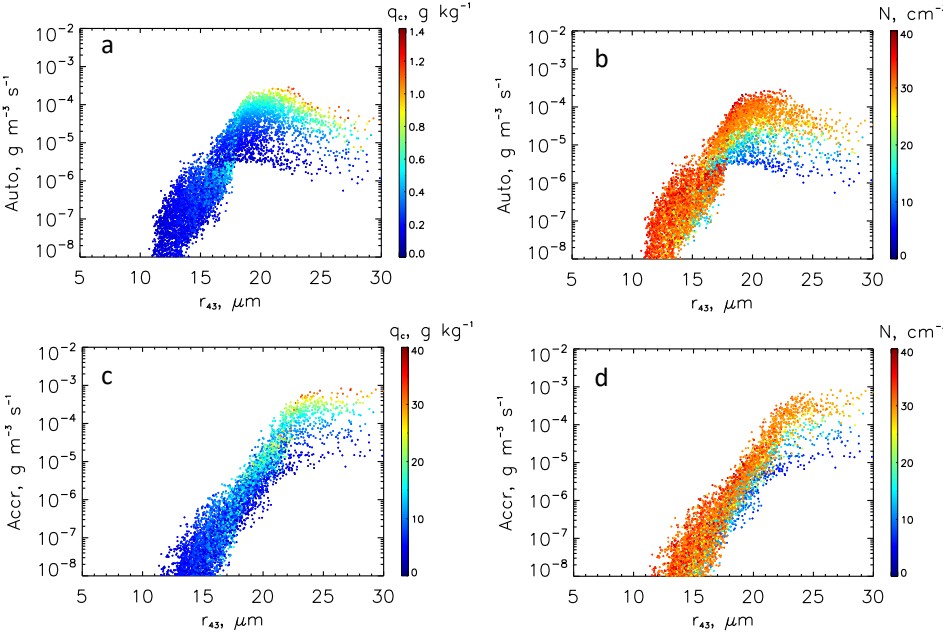

**Figure 5.** (a) and (b) autoconversion rates and (c) and (d) accretion rates as a function of $r_{43}$ color-coded by liquid water mixing ratio $q_c$ (a and c) and drop concentration $N$ (b and e). Note how autoconversion rates decrease for $r_{43} > 20\ \mu$m while accretion rates continue to rise. The relatively low values of autoconversion and accretion rates at high $r_{43}$ can be seen to be associated with co-occurrence of low $q_c$ and low $N$.



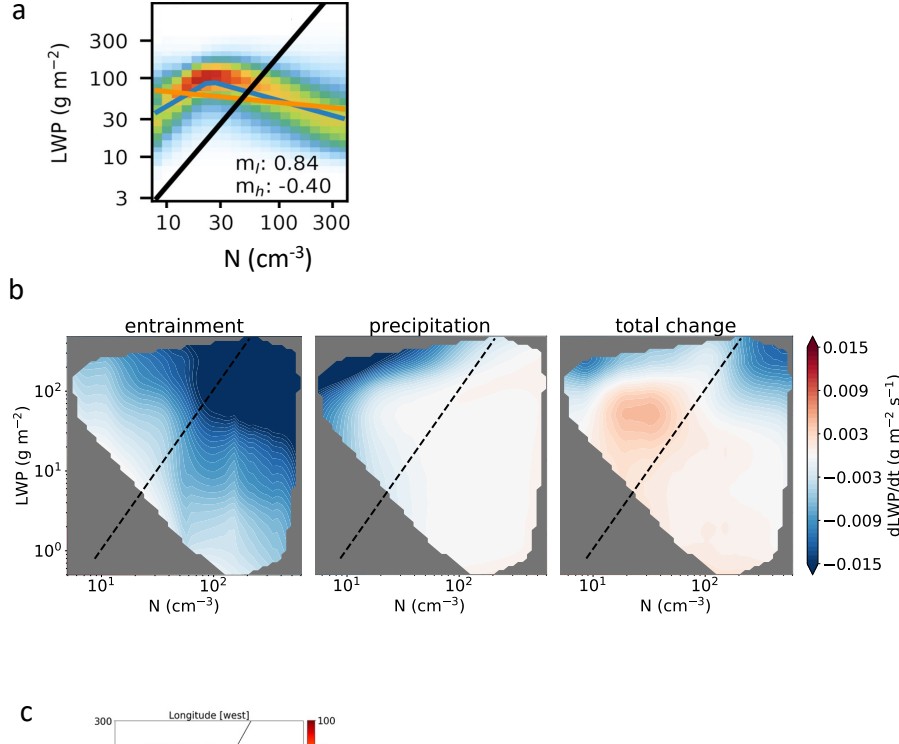

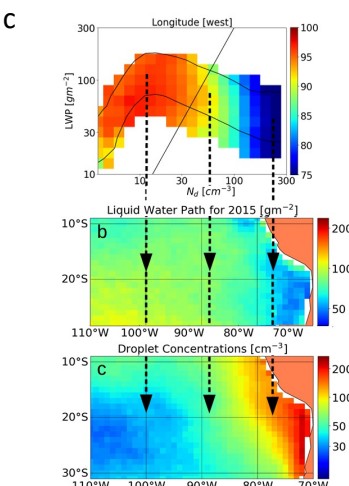

**Figure 6.** (a) Example of an LWP vs. N analysis based on MODIS data demonstrating an inverted V (Gryspeerdt et al. (2019), courtesy ACP). The positive and negative slopes are associated with precipitation suppression and evaporation feedbacks, respectively. The slopes associated with the inverted V (blue lines) are given as $m_l$ and $m_h$. The yellow line is a best fit line to all the data while the thick black line indicates an $r_e$ of 15 $\mu$m; (b) Connecting inverted Vs to processes. The figure shows LWP tendencies based on analysis of a large ensemble of LES. Blue regions indicate losses associated with precipitation and evaporation while the prominent red region is associated with very weak precipitation evaporating just below cloud based strengthening updrafts. (Adapted from Hoffmann et al. (2020), courtesy of the American Meteorological Society, with permission.); (c) LWP-$N$ analyses of the stratocumulus-to-cumulus transition after Goren et al. (2025, courtesy ACP) suggesting that inverted Vs are a consequence of large-scale boundary layer processes and co-variability of meteorology and aerosol. See text for further discussion.



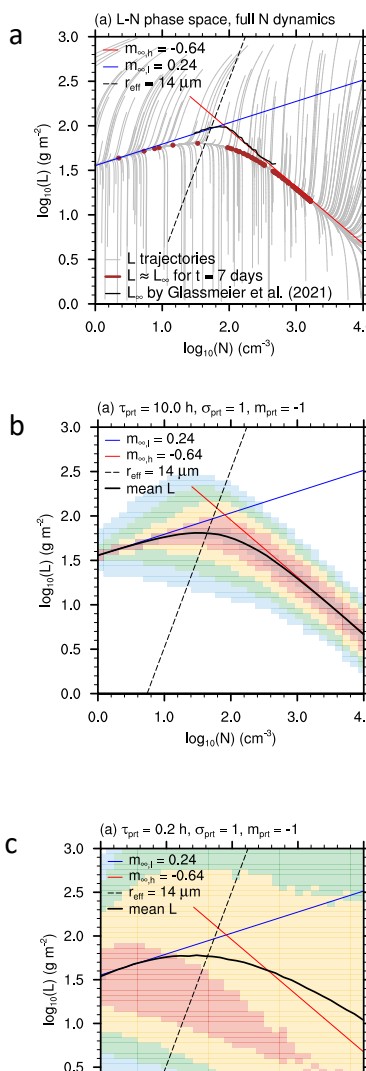

**Figure 7.** Heuristic model output exploring the effect of microphysical processes and LWP recharge on LWP vs. $N$; (a) multiple trajectories for different initial conditions converging to an inverted V (thin grey lines). Blue and red lines are based on the LES output of Glassmeier et al. (2021). The dashed line indicates an $r_e$ of 14 $\mu$m; (b) as in (a) but with long timescale external perturbations (Type 1) that still reproduce the essence of results in (a); (c) as in (b) but with short timescale external perturbations that significantly disrupt the shape of the inverted V (Type 2). In panels (b) and (c) the color shading reflects all the model output while the thick black line represents the mean. All panels derive from Hoffmann et al. (2024, courtesy ACP), with details furnished therein.



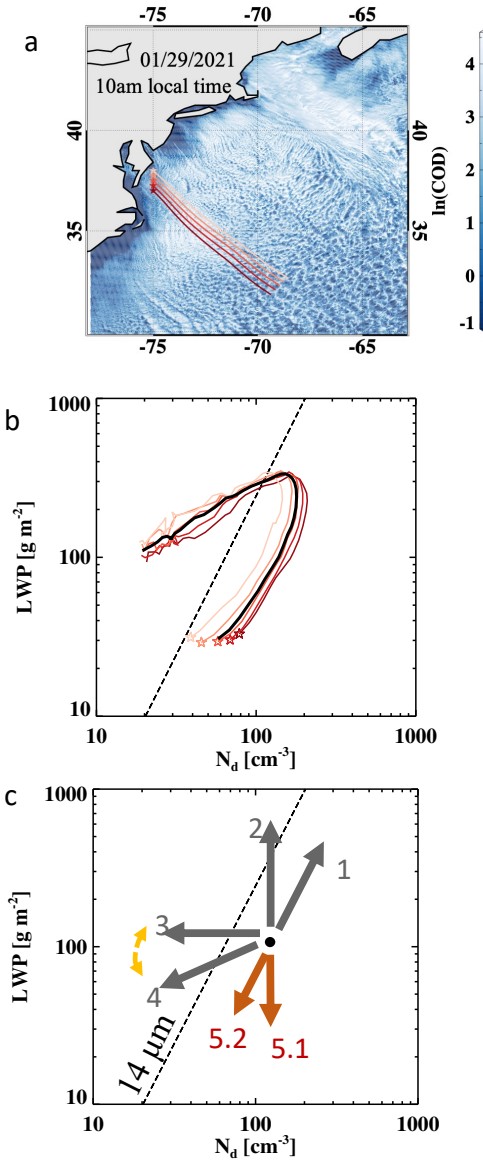

**Figure 8.** Analysis of a cold air outbreak demonstrating the ability to infer processes from GOES retrievals in a single satellite image together with information on the airflow through the image from ERA-5 reanalysis. (a) Snapshot of a cold-air outbreak scene sampled by GOES with superimposed instantaneous trajectories based on ERA-5 reanalysis; (b) Trace of the family of instantaneous trajectories through the scene in LWP-$N$ space; (c) A diagram identifying how individual processes drive the system in LWP-$N$ space (1. activation, 2. condensation, 3. collision-coalescence, 4. precipitation, 5. entrainment (5.1: homogeneous, 5.2: inhomogeneous)