# Peer review of "Opinion: Inferring Process from Snapshots of Cloud Systems"

_EGUsphere, 2025_

## Community Comment (CC1)

I found this article very interesting, and I have a couple of points for the authors to consider that I hope I have not expressed too incoherently.

First, I would like to push back against the language that features in the abstract and introduction stating that cloud and aerosol properties are averaged to 20 – 100 km resolution to reduce uncertainties. Averaging is not a strategy to deal with uncertainty. Averaging reduces random error, but any statistical estimation procedure can deal with such random errors, even if they are heteroskedastic across the samples/pixels. In fact, in the heteroskedastic case, simple averaging is not a good idea even if any subsequent inference is based entirely on areal averages. Beyond that, we know that remote sensing retrievals of cloud properties are not dominated by random error (e.g., radiometric noise). Instead, it is deterministic error due to algorithmic assumptions and finite resolution (i.e., errors are functions of the unknown cloud state) and spatially correlated errors in ancillary data that are the dominant sources of uncertainty. We would need to know how to model these deterministic errors with error covariances to propagate uncertainty. If we don't know how to model these errors, it doesn't matter whether we average or not, uncertainty is unspecified and the issue remains. So, the averaging is largely orthogonal to the matter of uncertainty and its propagation in inference. We have chosen to average; it is not a behavior that is prescribed by measurement limitations. Certainly, averaging doesn't meaningfully address any significant measurement limitations.

I suggest instead that conceptual simplification and practical issues of data size and computing power are the reason for averaging. As we have progressed in our understanding, we have moved from asking simple questions such as "What make more cloud?" to asking "What controls the intra-cell covariance of droplet concentration and liquid water path across closed-celled stratocumulus?" Answering the former question only requires coarse averages. As our theories grow in detail and subtlety we need to interpret our measurements with more nuance. This leads to a natural transition from only using coarse-resolution averages to analyzing the details of the spatial structure of cloud fields. I agree with the author's suggested direction; that there is much more to learn from snapshots when we interpret them correctly.

It appears to me that there is an assumed separation between the estimation of 'the rules of the game' and the knowledge that 'the rules are invariant' across a set of samples. To me, it is not clear that this is the case. When reading, I don't see a clear definition of which geophysical variables or properties we can use as evidence that 'the rules are invariant' and which we can use to determine the rules themselves (i.e., constrain processes).

To put this in more concrete terms, let's say that we want to perform a Bayesian inference of a vector of parameters of a microphysical parameterization, $\theta$ using a vector of observations $y$. We

need to consider the influence of the meteorological state, $\gamma$, on our observations so we must estimate the posterior distribution of these two sets of variables together:

$$p(\theta, \gamma | y) = \frac{p(y|\theta, \gamma) p(\theta, \gamma)}{p(y)} \tag{1}$$

What I believe is being assumed is that these kinds of estimation problems are separable so that

$$p(\theta, \gamma | y) \approx p(\theta | y_1) p(\gamma, y_2) = \frac{p(y_1|\theta) p(\theta)}{p(y_1)} \frac{p(y_2|\gamma) p(\gamma)}{p(y_2)} \tag{2}$$

where $y_1$ and $y_2$ are non-overlapping subsets of observations.

For example, when we perform a reanalysis, we estimate large-scale thermodynamic and dynamic properties such as winds, temperature and humidity using measurements of those same properties, $y_2$, but not measurements of cloud microphysics ($y_1$), and we do not jointly estimate the microphysical parameters, $\theta$. So, we obtain a maximum a posteriori estimate of the meteorological state (i.e., reanalysis):

$$\tilde{\gamma} = \underset{\gamma}{\mathrm{argmax}}\, p(\gamma | y_2) \tag{3}$$

Then, we use measurements of cloud properties, to estimate parameters of cloud processes conditioned on knowledge of the meteorological state from reanalysis:

$$p(\theta | y_1, \tilde{\gamma}) \frac{p(y_1|\theta, \tilde{\gamma}) p(\theta)}{p(y_1)} \tag{4}$$

In Eq. 4 we find the 'rules of the game', $\theta$, from observations under the assumption that the rules are invariant (fixed $\tilde{\gamma}$).

For example, for the stratocumulus case, it is stated that the inversion height is horizontally homogeneous, so the 'rules are invariant', and yet variation in the inversion height appears to be integral to the intra-cell variability as well from Fig. 2. Is there a clear way to identify this separability? Again, for the stratocumulus, do we know a priori that there are no drivers that operate at scales between the cellular scale and ~100 km? Or are we relying on observations (reanalysis?) that demonstrate a lack of variance at this range of scales? For the cold-air outbreak trajectory example, the timescales discussed only mentions the timescale of SST gradient. Could there not be meteorological changes that are significant at a timescale of ~12 hours associated with synoptic systems that cold-air outbreaks are often part of?

I think it would be great if the authors could be a bit more precise in how they would determine that the 'rules are invariant'. Eq. 4 seems to be assumed ubiquitously. For example, we assume we do not need to solve a data assimilation problem to calibrate a climate model. In other words, we assume that "we don't need to know the weather to project the climate", though I've yet to see any

evidence of this. Is this separability real or do we impose (assume?) a scale-break between the resolution of global reanalysis/climate model and the domain size of Large Eddy Simulations that is just an artifact of computational limitations? This is a critical assumption that also appears to underly the authors' arguments, so it would be great to get their opinion on it.

The authors' notion of how processes can be inferred from snapshots is more specific than my Eq. 4, arguing that, for Type 1 systems, there is information about a fast microphysical process ($\theta$) even when $y_1$ are effectively contemporaneous. Am I correct in understanding that ergodicity implies that we can interpret the droplet effective radius profile in cumulus or the cellular structure in closed-celled stratocumulus using a parcel model, rather than requiring a whole LES? If this is true, it becomes a lot simpler (especially computationally) to evaluate the likelihood, $p(y_1|\theta, \tilde{\gamma})$, and the corresponding posterior. If this is the case, it would be good for the authors to state it explicitly or if not, it would be good to have a more practical statement of how exactly ergodicity would simplify inference of processes. Even in the effective radius profiling example, there is a clear influence of thermodynamics through condensation rates etc., and again the issue of separability in the inference of a microphysical process arises. I agree that developing a deep understanding the mechanism for the apparent ergodicity is extremely important to justify this sort of observational interpretation.

The notion that processes can be extracted from Type 1 snapshots suggests to me that we might get more value from observing systems that provide high detail and accuracy in select conditions (at the expense of sparse sampling) rather than those that sample everything but with little detail or precision. Does this align with the authors' understanding? If so, it might be worth making a recommendation along those lines.

Perhaps I am misunderstanding, but I have some concerns about the Type 2 cases, where it is stated that they may be useful after careful stratification by meteorology. If drivers such as aerosol and 'meteorology' are correlated across snapshots, then stratification (or other statistical models and their counterfactuals) will produce biased estimates of how clouds respond to an aerosol driver under 'constant meteorology' (and vice versa). The assumption that including a variable as a covariate in a statistical model will control for its influence is a fallacy. For example, if the mechanism by which aerosol affects cloud fraction is changes in sub-cloud stability from precipitation suppression, then including a variable that correlates with any of the variables involved in this process as a covariate in a statistical model or as a stratification parameter will bias the apparent susceptibility (i.e., partial derivative) of cloud fraction to aerosol with respect to a true causal response. Statistical frameworks for analysing 'causality' can only untangle these

effects when processes are resolved by the measurements, i.e., not Type 2, and all relevant variables in the causal graph are observed.

The stratification approach seems to underpin the entirety of ACI analysis. It is unsurprising then that direct estimates of things like radiative forcing due to ACI from statistical counterfactuals seem highly disconnected from the actual large-scale behavior and processes, as recently found for liquid water path adjustments. To borrow the statistical terminology of medicine, cross-sectional statistics (like sets of snapshots) are not definitive but can motivate proper study design (e.g., randomization) to estimate effect sizes. Even in a longitudinal study (i.e., tracking clouds with geostationary satellites), there is difficulty in untangling causality.

I would suggest that direct estimation of 'processes' from statistical models built on Type 2 observations is not going to be robust, as 'drivers' will also be correlated with intermediate variables. Instead, I would advocate for an observation-constrained model-based counterfactual in which Eq. 4 (or better Eq. 1) is evaluated and then the calibrated model is used to compute a counterfactual. In other words, I am arguing that Type 2 cases do not provide a shortcut to access process understanding. I think it would be helpful for the authors to be a bit more precise about the conditions required for Type 2 cases to be helpful for process understanding in terms of controlling for the variation of slow processes across snapshots.

I strongly support the closing statement that existing measurement limitations should not get in the way of the refinement of conceptual thinking regarding how we can best extract understanding from measurements. In fact, I hope that refining our conceptual thinking will drive innovation in measurement. For example, currently satellite remote sensing measurements formulate their scientific accuracy requirements without any consideration of spatial error covariance (This is the cause of the uncertainty issue discussed first). If we show that process understanding comes from spatial patterns in snapshots, then the requirements should be set in terms of spatial error covariance. Confronting our existing algorithms with such a requirement will drive innovation.

The authors make the statement:

Line 401-402: "At high solar zenith angles (SZA) retrievals are more problematic but events that lie within the optimal SZA window (less than 65°; Grosvenor et al., 2018) will be valuable."

This statement that events within the SZA < 65 are optimal and therefore valuable for studying cloud processes is not consistent with the available evidence.

Here, the claim that SZA > 65 are insufficiently accurate is conflated with the claim that retrievals with SZA < 65 are sufficiently accurate. A reading of Grosvenor et al. 2018 and the references within reveals that the sufficiency of operational geostationary retrievals with SZA < 65 to be

'valuable' for studying the covariance of droplet concentration and liquid water path etc. has never been demonstrated. The conflation of these points is a widespread fallacy in the use of satellite remote sensing data to study aerosol cloud interactions, i.e., "we excluded the lowest-quality data so now what we have left are good quality" (not just less low-quality). See Loveridge & Di Girolamo (2024) for more discussion of this point.

I suggest that the authors simply follow the spirit of their closing statement and avoid distracting from their main point by discussing details of measurement performance. The main point of this paragraph, that measurements with wide field of view and high temporal frequency will be useful, has the same caveat as all measurements (sufficient accuracy) that are discussed in the article. I don't think the authors should stress over justifying this particular type of measurement.

References:

Loveridge, J. R., & Di Girolamo, L. (2024). Do subsampling strategies reduce the confounding effect of errors in bispectral retrievals on estimates of aerosol cloud interactions? *Journal of Geophysical Research: Atmospheres*, 129, e2023JD040189. https://doi.org/10.1029/2023JD040189

---

## Author Comment (AC1)

**Response to Reviewer 2**

General comment:

This paper provides a nice overview of how to obtain process information from snapshot measurements of cloud systems through reviewing the literature of previous observational and modeling studies. The previous relevant works are reviewed in the well-organized manner that classifies the past approaches according to relative magnitudes of time scales of phenomena and observations. I think that this review is enlightening and encouraging to further explore key fundamental processes of cloud systems through deliberate use of measurement data to obtain observation-based process understanding that is highly required these days to essentially advance numerical modeling of clouds. I only have a couple of minor comments listed below that I hope can be addressed easily by the authors. After the authors address those points, I recommend this manuscript to be published.

We thank the reviewer for their positive comments!

Specific points:

Line 74-75: It is a bit unclear to me what the "observation timescale" means. At first, I interpreted it to be a "temporal resolution" of observation, but later I realized that this means what is more like a "duration of observation". Is this interpretation correct? I would appreciate the authors to clarify this point to avoid possible confusion for interpreting the meaning of the Deborah number and the classification into Type 1 and 2.

The reviewer is correct: the observation timescale is the duration of observation. We now make this clear in the revised manuscript on first usage on line 74.

Section 2: Given a remarkable progress in satellite observations with active sensors in this couple of decades, I'm just curious how various types of statistical analysis with vertical profiling data from radar/lidar are classified into the two types the authors defined. In particular, I'm wondering how three statistical methods of compositing the vertical cloud profiles, namely, Contoured Frequency by Altitude Diagram (CFAD), Contoured Frequency by Temperature Diagram (CFED), and Contoured Frequency by Optical Depth Diagram (CFODD), are classified into the two types or any other type. A-Train satellite data is touched on in Section 4, but more detailed discussion of active sensor-based analysis would be appreciated.

Statistical compositing methods can be applied to single storms or to large composites. Based on the ideas laid out here, the former is likely to yield better physical constraints than the latter because of the increasing likelihood of changing conditions with multi-day composites. We expect that changing conditions would generate more variance in e.g., CFODD plots.

Note that one of the first examples we introduced in the original was for surface radar tracking a storm system over its lifetime (lines 77-80 in the revised manuscript) but we now add more

text on statistical compositing as in work by e.g., Suzuki et al. (2010, DOI: 10.1175/2010JAS3463.1). In keeping with our discussion of the Stephens and Haynes example, we now add the following text on lines 262-266:

*A related topic is the use of space-based radar and spectrometer retrievals of Z and COD, respectively, to interpret the relative importance of condensation growth (higher COD but almost no change in Z) and collision-coalescence growth (higher Z but little to no change in COD) (Suzuki et al., 2010). Based on the arguments above, when applied to single storm systems one expects such data to be of Type 1, but when compositing over many storms with different dynamics the analysis is expected to be of Type 2.*

Section 2.3: As a quick note on Stephens and Haynes (2007), I like to point out that the left-hand side and right-hand side of equation (2) are not obtained from independent measurement information. The left-hand side quantity (P times h) is derived from $r_e$, COD and Z, according to the expression on the right-hand side. By carefully looking at the right-hand side, the timescale of auto-conversion, represented by the slope in Figure 3, is solely determined by Z. This correspondence of Z to the timescale comes from the assumption of Long's collection kernel proportional to sixth power of particle radius that happens to coincide with the dependence of Z on particle radius (which is also sixth power). Constrained by this assumption, the variability range of the timescale (or slope in Figure 3) simply reflects the variability range of Z bracketed between -15dBZ and 0dBZ. This understanding of Stephens and Haynes (2007) should be more clearly described in the authors' argument of Line 224-230 to interpret "why the process rates are relatively poorly constrained". Again, the diversity of the timescale is just a simple translation from the diversity of Z, according to equation (2).

The reviewer is correct. We have modified the text to clarify the Stephens and Haynes methodology and the origin of Z on the RHS of the equation. We still keep this brief in order to focus on the conceptual aspects of the paper. See changes on lines

230-231: *Of note is that the appearance of $\bar Z$ in Eq. (2) derives from Long's collection kernel for small drops, which has an $r^6$ dependency.*

238-239: *Because the kernel function is a function of $r^6$, the range of time-scales simply reflects the variability in $\bar Z$ bracketed between -15 dBZ and 0 dBZ.*

---

## Author Comment (AC2)

**Response to Reviewer 1**

This manuscript delivers an elegant and much-needed synthesis of how and when snapshot observations of clouds can justifiably be interpreted as proxies for time-resolved processes. Its intellectual clarity and breadth of examples promise to influence both observationalists and modelers.

Inferring process from snapshots of cloud systems is a thought-provoking synthesis: it distils scattered intuitions about when spatial statistics can stand in for temporal evolution, and it offers a clear vocabulary (ergodicity, D-number, Type 1 vs. Type 2) that would help future cloud research.

In my opinion, apart from a few minor modifications, the paper should be published. It is an "opinion" paper, and as such, it highlights an important question that is highly relevant to cloud/rain/aerosol/climate research in a rather qualitative manner.

We thank the reviewer for their positive comments!

Minor comments:

1) In general, I miss a consideration of the variance of the explored processes. In all of the examples, the mapping is not one-to-one. The relevant variables are represented by a distribution (r_e or LWP). When sampling the state space, one must be sure that the variability is covered.

The variance is not necessarily a reflection of the many states. It could reflect variations around a given state. The text in Line 141, for example. The step of translating the satellite snapshot into a few hours of observation is critical. It distills the essence of the paper and should be better explained. We know that the r_e slices (per T, Z, or P) can be highly variable. I miss a discussion of the need for fully covering the statistical variance.

The question is a good one. We will distinguish between three different sources of variance: i) variance caused by varying meteorological conditions ("varying rules of the game", Type 2; ii) variance in cloud top $r_e(z)$ across the domain, which is required for the cloud-scale profiling to be effective; and (iii) variance rooted in cloud-scale fluctuations, especially from turbulent mixing, which is not the scale of interest in the problem posed here. For our targeted cloud-scale process timescale, in order to capture this variance and obtain sufficiently precise mean values, one has to sample a sufficient number of clouds over the range of z of interest. The sample size will vary with altitude, there being far more small/shallow clouds than large/deep clouds. To improve statistics, one would like to have a large enough domain with the proviso that there is no significant change in the meteorological state across the domain.

The LES results represent a very large statistical sample of clouds and remarkably show a tight $r_e(z)$.

In the interests of keeping the manuscript conceptual, we have not attempted to quantify variance that might derive from process changes (e.g. fluctuations in entrainment), meteorological changes (large scale, or perhaps associated with clouds themselves affecting their local environment), or from sampling statistics. We have, however, added clarifying comments to the text, particularly regarding the scale separation inherent to the different examples.

The text has been modified on lines 152-157:

*When applying this method to satellite-based observations one has to sample a sufficient number of clouds over the range of heights of interest. The sample size will vary with altitude, there being far more small/shallow clouds than large/deep clouds. To improve statistics, one has to balance the desire for a large domain with the risk of gradients in the meteorological state across the domain. We have not attempted to quantify the variance associated with small-scale process changes (e.g. fluctuations in entrainment-mixing), meteorological changes (large scale, or perhaps associated with clouds themselves affecting their local environment), or sampling statistics.*

2) On the same note, what is a sufficiently large snapshot? How to scale the spatial length of it to the time it covers? What is the right mapping constant? Is it advection?

As noted above, a sufficiently large snapshot would be one in which one has a sufficient number of clouds over the range of z of interest the domain is large enough that there is no significant change in the meteorological state across the domain. The overall time duration for this case is connected to the range of cloud heights that the cloud system spans rather than an advective timescale.

See text on lines 161-163: "*The relevant process timescale $\tau_{proc}$ is associated with the evolution of an individual cloud or, more specifically, the height increment between the individual clouds, ordered by height (say 100 m). For typical updrafts on the order of meters per second $\tau_{proc}$ would be on the order of minutes.*"

3) Line 237: "Because the data derive from many different conditions, the observation timescale t_obs is on the order of many days …" Please explain why, when mixing many observations of different thermodynamical states, we can scale the observation time to days? I guess that by doing so, we average over many thermodynamic scenarios? Again, in this case, the variance of the timescale is important.

First, increasing the number of days included in the analysis naturally corresponds to a large observation timescale because many different cloud and environmental states are being observed. However, the crux here is that when aggregating data from many days one encompasses many cloud and environmental conditions. Individual systems therefore

experience different environments and therefore respond differently, obfuscating the underlying processes of interest, leading to a Type-2 designation.

See text on lines 253-257:
*The large number of days included in the analysis corresponds to a large observation timescale because many different environmental states--which translate to different cloud dynamical states--are being observed. The changing dynamical state is important because it brings in processes such as droplet nucleation, in-cloud residence times, and fallout, which could obscure the collision-coalescence process.*

4) What is the meaning of ergodicity in the case of averaging many states? I think that discussing it in the introduction would be beneficial to the general message of the paper. Can any system be averaged such that taking enough samples to cover the state distribution will yield an ergodic system?

Averaging over many systems that evolve under different states can be considered equivalent to a very long timeseries that explores all these states over time. Averaging over many states thus increases the observational timescale and accordingly means that slower processes are sampled.  As one does so, one increases the range of spatial/temporal scales that are included in the analysis, leading to less and less useful results. For example, if one were to average $r_e(z)$ over a very large area—say global, just to make a point—one would obtain an $r_e(z)$ profile that is representative of global conditions but has very little relevance to microphysics. In other words, with progressive averaging one loses the timescale of interest until the result isn't very useful because it encompasses a range of scales large enough that one cannot learn anything specific about the target (small-scale) process.

Regarding the latter part of this question, the answer is 'no' because the rules keep changing and the range of processes keeps increasing.

---

## Author Comment (AC3)

**Response to Comment CC1**

We thank Dr. Loveridge for his interesting perspectives. We extract below key themes from his comment and respond to each in turn

1) First, I would like to push back against the language that features in the abstract and introduction stating that cloud and aerosol properties are averaged to 20 – 100 km resolution to reduce uncertainties. Averaging is not a strategy to deal with uncertainty.

   Thank you for this comment. We agree and now reword the text to focus on the practical reasons for averaging in the context of our analysis. (lines 6, 25, 116)

2) It appears to me that there is an assumed separation between the estimation of 'the rules of the game' and the knowledge that 'the rules are invariant' across a set of samples. To me, it is not clear that this is the case. When reading, I don't see a clear definition of which geophysical variables or properties we can use as evidence that 'the rules are invariant' and which we can use to determine the rules themselves (i.e., constrain processes).

   First, we would like to clarify that the 'rules of the game' are not being estimated. We are attempting to understand processes amongst geophysical variables within systems that may be experiencing meteorological changes over the course of the observation period, i.e., changing rules of the game.
   We agree that this perspective assumes a timescale separation between the variability of the meteorological conditions and the process(es), which we are studying. This assumption is implicit in our definition of Type 1 and Type 2, which are both based on the process timescale being much smaller than the meteorological timescale. While this might not always be the case, one might argue that this assumption may be sufficiently fulfilled in the successful examples of space-time exchange covered in the manuscript. We will provide further justification for this assumption below.

   We now mention the timescale and associated process scale separation in the paper Lines 206-207: *Lending success to this approach is that there is sufficient scale separation between mesoscale processes (our focus), small-scale processes such as cloud-top entrainment or local plume penetration, and longer timescale variability inversion height.*

3) For example, for the stratocumulus case, it is stated that the inversion height is horizontally homogeneous, so the 'rules are invariant', and yet variation in the inversion height appears to be integral to the intra-cell variability as well from Fig. 2.

Broadly speaking stratocumulus (Sc) are characterized by relatively flat tops and ragged bases – as opposed to cumulus with flat bases and highly variable cloud tops. The 'flat-topped Sc are a consequence of a strong inversion. In Lilly's mixed-layer model of the stratocumulus-topped boundary layer, the evolution of this inversion height has a characteristic timescale of 48 hours, while the thermodynamic properties of the boundary layer evolve on a shorter timescale of about 9 hours (Schubert et al., 1979). To the extent that 9 hours can be considered "much smaller" than 49 hours, this justifies an assumed timescale separation.

Note too that we are referring to the *domain-mean* inversion height. Of course, there are local variations in the inversion height, but they are part of another set of rules and timescales (Rayleigh-Benard convection), which are not the focus here. Thus, we select our methodology to focus on processes/timescales of interest.

Changes have been made on lines 203-204

4) Again, for the stratocumulus, do we know a priori that there are no drivers that operate at scales between the cellular scale and ~100 km? Or are we relying on observations (reanalysis?) that demonstrate a lack of variance at this range of scales?

Our analysis of the Sc cellular structure is an example of how observational evidence (cell size, cell aspect ratio, radiative cooling, circulation) can be used along with detailed modeling to build a composite characteristic cell comprising samples from many different cells collected into TWP bins. If meteorological conditions vary across the domain – e.g., changing inversion strength – one might find that various other scales emerge. Our intent has simply been to show that if meteorological conditions are reasonably constant, Sc cells manifest with remarkable self-similarity. This seems in line with assuming timescale separation as discussed above.

5) For the cold-air outbreak trajectory example, the timescales discussed only mentions the timescale of SST gradient. Could there not be meteorological changes that are significant at a timescale of ~12 hours associated with synoptic systems that cold-air outbreaks are often part of?

There are no doubt other meteorological changes that do occur, which will vary from one meteorological state to another. Having looked at a large number of MCAO cases we have seen that sometimes the meteorological gradient *does* change within ~12 h. Nevertheless, SST is a very strong controlling factor and as shown in the analysis, there

are indications that one can learn about cloud processes based on our analysis approach. This is again in line with our previous arguments for assuming a timescale separation for the stratocumulus-topped boundary layer.

6) I think it would be great if the authors could be a bit more precise in how they would determine that the 'rules are invariant'.

We pay more attention to this in the revised version while keeping the manuscript conceptual and focused on the key theme of space-time exchange, Deborah number, and ergodicity.

7) Is this separability real or do we impose (assume?) a scale-break between the resolution of global reanalysis/climate model and the domain size of Large Eddy Simulations that is just an artifact of computational limitations? This is a critical assumption that also appears to underly the authors' arguments, so it would be great to get their opinion on it.

Global models are good at representing large scale phenomena that LES with their limited domains cannot capture. Likewise, LES captures small-scale physical processes that global models cannot. Modelers always deal with the issue of scale-filtering whereas clearly the atmosphere does no such thing. This doesn't mean, however, that models of a certain filter scale are not useful for a selected problem. The examples presented here include a mix of complementary modeling and observational studies and demonstrate that in ideal cases – particularly Type 1 where meteorological conditions are 'invariant'– much can be learned from snapshots, and that assuming scale separation can be a useful assumption.

8) Am I correct in understanding that ergodicity implies that we can interpret the droplet effective radius profile in cumulus or the cellular structure in closed-celled stratocumulus using a parcel model, rather than requiring a whole LES?

No, not directly because the $r_e$ profile depends on the dynamics of the cloud system and details of entrainment. A parcel model typically does not represent entrainment, and when it does it is heavily parameterized. It could however be argued that ergodicity implies that it not fundamentally impossible – for given meteorological conditions and potentially very complex sets of parameters – to parameterize cloud evolution as simulated by an LES through a parcel model.

9) The notion that processes can be extracted from Type 1 snapshots suggests to me that we might get more value from observing systems that provide high detail and accuracy in

select conditions (at the expense of sparse sampling) rather than those that sample everything but with little detail or precision. Does this align with the authors' understanding? If so, it might be worth making a recommendation along those lines.

There are multiple aspects to this question including what geophysical variables/processes are being targeted for which scientific question and what observational systems are being considered. Even though certainly of interest, we feel that this question is beyond the scope of the current paper.

10) Perhaps I am misunderstanding, but I have some concerns about the Type 2 cases, where it is stated that they may be useful after careful stratification by meteorology. If drivers such as aerosol and 'meteorology' are correlated across snapshots, then stratification (or other statistical models and their counterfactuals) will produce biased estimates of how clouds respond to an aerosol driver under 'constant meteorology' (and vice versa).

This is true when one only considers a specific meteorological regime. However, even if meteorology and aerosol are correlated, one can obtain causal understanding under those specific meteorological conditions. One can then collect many such covarying conditions. Finally, one can scale up by taking into account of the frequency of occurrence of all these conditions. In this case we argue that the scaled-up understanding is not biased (e.g., https://doi.org/10.5194/acp-22-861-2022).

11) I think it would be helpful for the authors to be a bit more precise about the conditions required for Type 2 cases to be helpful for process understanding in terms of controlling for the variation of slow processes across snapshots.

We have attempted to do so for the examples given here but now add more discussion. For example, we now put more focus on the role of dynamics in the Stephens and Haynes Type 2 retrieval and its connections to the Z-COD composites of Suzuki et al. (2010).

*Lines 262-266: A related topic is the use of space-based radar and spectrometer retrievals of $Z$ and COD, respectively, to interpret the relative importance of condensation growth (higher COD but almost no change in $Z$) and collision-coalescence growth (higher $Z$ but little to no change in COD) \citep{Suzuki10}. Based on the arguments above, when applied to single storm systems one expects such data to be of Type 1, but when compositing over many storms with different dynamics the analysis is expected to be of Type 2.*

12) As an opinion piece, our intent is to provide food for thought with a number of specific examples. A comprehensive discussion lies beyond our scope.

13) This statement that events within the SZA < 65 are optimal and therefore valuable for studying cloud processes is not consistent with the available evidence.

Thank you for clarifying this important point. SZA<65 was recommended by Grosvenor et al. 2018 to filter out highly uncertain Nd retrievals but this doesn't mean that after this screening the retrievals are free of uncertainty. We have made the appropriate changes on lines 425-426: *This approach would have to take into account uncertainties in retrievals, particularly at high solar zenith angles \citep[e.g.,][]{Grosvenor18}.*

14) I suggest that the authors simply follow the spirit of their closing statement and avoid distracting from their main point by discussing details of measurement performance. The main point of this paragraph, that measurements with wide field of view and high temporal frequency will be useful, has the same caveat as all measurements (sufficient accuracy) that are discussed in the article. I don't think the authors should stress over justifying this particular type of measurement.

Given that temporal evolution is inherent to process, we consider it important to emphasize temporally evolving (geostationary) data in trying to address process. Measurement inaccuracies from the suite of instruments involved are certainly an issue.